# A Nonlinear Inexact Two-Stage Management Model for Agricultural Water Allocation under Uncertainty Based on the Heihe River Water Diversion Plan

**DOI:** 10.3390/ijerph16111884

**Published:** 2019-05-28

**Authors:** Chenglong Zhang, Qiong Yue, Ping Guo

**Affiliations:** Center for Agricultural Water Research in China, College of Water Resources and Civil Engineering, China Agricultural University, Beijing 100083, China; zhangcl1992@cau.edu.cn (C.Z.); yueqxy@163.com (Q.Y.)

**Keywords:** two-stage stochastic programming, nonlinear objective, agricultural water allocation, Heihe River ecological water diversion plan, interval regression analysis

## Abstract

In this study, a nonlinear inexact two-stage management (NITM) model is proposed for optimal agricultural irrigation water management problems under uncertainty conditions. The model is derived from incorporating interval parameter programming (IPP), two-stage stochastic programming (TSP) and quadratic programming (QP) within the agricultural water management model. This model simultaneously handles uncertainties not only in discrete intervals, but also in probability distributions, as well as nonlinearity in the objective function. A concept of the law of diminishing marginal utility is introduced to reflect the relationship between unit benefits and allocated water, which can overcome the limitation of general TSP framework with a linear objective function. Moreover, these inexact linear functions of allocated water can be obtained by an interval regression analysis method. The model is applied to a real-world case study for optimal irrigation water allocation in midstream area of the Heihe River Basin in northwest China. Two Heihe River ecological water diversion plans, i.e., the original plan and an improved plan, will be used to determine the surface water availabilities under different inflow levels. Four scenarios associated with different irrigation target settings are examined. The results show that the entire study system can arrive at a minimum marginal utility and obtain maximum system benefits when optimal irrigation water allocations are the deterministic values. Under the same inflow level, the improved plan leads to a lower water shortage level than that of the original plan, and thus leads to less system-failure risk level. Moreover, the growth rate of the upper bound of economic benefits between each of two scenarios based on the improved plan are greater than that from the original plan. Therefore, these obtained solutions can provide the basis of decision-making for agricultural water allocation under uncertainty.

## 1. Introduction

The Heihe River Basin (HRB) is the second largest inland river basin in an arid region of northwest China [1,2]. It is divided by the Yingluoxia and Zhengyixia hydrological stations into three segments. Both the midstream and downstream areas of the HRB act as the main water consumption areas. The midstream region especially is a major grain production base in China that currently faces severe water shortages [3]. In the past decades, agricultural water consumption in the midstream section and ecological water use in the downstream one are in competition. This results in a reduction in downstream ecological water use, thereby leading to the deterioration of ecological sustainability. Consequently, in the 1990s, the central government put forward the Heihe River ecological water diversion plan (i.e., “97” plan, original plan) to alleviate the ecological deterioration [4]. That is, the water released to the downstream from the Zhengyixia hydrological station should be guaranteed based on the runoff from the Yingluoxia hydrological station under different inflow levels. However, based on the recent cases of water released, there are several shortcomings during the actual application [4,5]. For example, the released targets were set much higher assuming a high inflow level so that it usually couldn’t fulfill the release requirements. Moreover, the low precipitation levels and the implemented ecological water diversion plan jointly diminished the midstream surface water availability. Furthermore, more than 90% of the total water consumption of the midstream area is for agricultural production [6]. Increasing agricultural water demand and diminishing water supply has led to dramatic conflicts among multiple competing water users throughout the HRB. This further causes increased groundwater exploitation resulting in falling water tables and unsustainable development of agricultural production. Therefore, effective agricultural water resource management will be beneficial not only for agricultural production and food security, but also groundwater protection and ecological development. 

Previously, many system analysis methods have been developed for agricultural water management problems with limited water resources [7,8,9,10,11,12,13,14,15,16]. However, the above traditional methods optimization models can hardly handle various uncertainties in the agricultural irrigation water management problems. For example, accurate quantitative data on parameters such as runoff inflows, water availability, irrigation targets, irrigation quota, and market prices are not easy to obtain [17]. Furthermore, interrelationships among these factors render problems more complicated, and thus affect economic implications. Therefore, in response to those uncertainties existing in agricultural system, introducing uncertainty theory into traditional optimization method to handle various uncertain factors and their interrelationships will be an effective way to reflect the complexity of agricultural water management problems.

To address the above concerns, many inexact mathematical programming methods were developed to support water resources management [2,5,6,18,19,20,21,22,23,24,25,26,27,28]. Among them, two-stage stochastic programming (TSP) is effective for addressing problems associated with policy scenario analysis and uncertain random variables in models. In TSP, a decision is first given before random variables are known. When random events have happened, it allows corrective measures to be undertaken, and this is called the second decision. However, the quality of collected data is usually not good enough to be expressed as probability distributions or fuzzy membership function. Such uncertainties fluctuate within a relevant range between lower and upper bounds. Therefore, interval parameter programming can be incorporated into the TSP optimization framework [19]. Nevertheless, the above interval two-stage stochastic programming (ITSP) method is usually based on the assumption of a linear objective function, and it often ignores the effect of marginal utility. Thus, ITSP has difficulty in dealing with the effects of marginal utility between unit benefit and demand.

In the context of economics, a meaningful concept is the law of diminishing marginal utility, which refers to the fall in marginal utility as consumption increases. Similarly, the effects of marginal utility in fact may be significant when there is limited water available to multiple competing users, which will affect benefit or cost coefficients. Moreover, marginal utility effects are identifiable when analyzing the relationship between unit benefit and the amount of irrigation water. The unit benefit is defined as economic benefits per unit of the irrigation amount, which can be interpreted as economic water productivity for agricultural water allocation optimization system [27]. Therefore, based on the law of diminishing marginal utility, the unit benefit and penalty coefficients can be expressed as linear functions of the amount of allocated water and water shortages. By concentrating on the marginal utility of water, the system optimization model becomes a hydro-economic model. Hydro-economic models can incorporate subsystems including hydrologic, engineering, environmental and economic aspect into a water resources system within a general framework, which are solution-oriented methods to develop novel solutions for optimal water use [29]. Previous hydro-economics models have been investigated [30,31,32,33,34,35,36,37], but the majority of them can hardly address system uncertainties. For better reflecting actual conditions and uncertain information in study system, an interval regression analysis method will be introduced to calculate inexact linear functions of allocated water (e.g., $a_{j}^{\pm }x_{j}^{\pm }+b_{j}^{\pm }$, where the superscripts ‘-’ and ‘+’ denote the lower and upper bounds of an interval parameter/variable) instead of deterministic ones. Therefore, when the objective function is the maximum of economic benefits, nonlinear (quadratic) problems need to be addressed (e.g., $f^{\pm }=\left(a_{j}^{\pm }x_{j}^{\pm }+b_{j}^{\pm }\right)x_{j}^{\pm }$).

Quadratic programming (QP) can handle nonlinearity in the objective function, and achieve global-optimum acquisition under certain system conditions [24,38]. For example, Chen and Huang [39] incorporated interval parameter programming into a QP context to deal with uncertainty without specific probability characteristics, and proposed a derivative algorithm method to solve the inexact QP model more efficiently. Gu and Victor [40] presented a solution path algorithm for general parametric QP problem. However, it is hard to deal with random uncertainty, and it cannot present a linkage between economic implications and the violation of policy targets due to water shortages. Furthermore, few applications of the QP to agricultural water management have been undertaken.

Therefore, to better account for policy scenario analysis, the effects of marginal utility and nonlinearity in agricultural water management problems, a nonlinear inexact two-stage management (NITM) model is proposed under uncertainty based on the Heihe River water diversion plan. This model incorporates interval parameter programming and quadratic programming into TSP optimization framework. It handles uncertainties presented as both discrete intervals and probability distributions, as well as nonlinearity in the objective function. It overcomes the limitation of the TSP framework by introducing the concept of diminishing marginal utility. Thus, the model can reflect the effects of marginal utility between unit benefit and the amount of allocated irrigation water due to inexact linear functions of allocated water and shortage. Then, this model is applied to a real-world case study for supporting agricultural water management in midstream area of the HRB, an arid area in northwest China. Four scenarios associated with different irrigation target settings are examined. In detail, this study is organized as follows: (1) modifying the original water diversion plan to satisfy the requirements of the released water; (2) formulating the NITM model; (3) applying the proposed model to a case study and (4) analyzing the obtained results under different scenarios for comparisons.

## 2. Methodology

The methodology section consists of the following three subsections: (1) interval regression analysis method to obtain the interval linear functions of allocated water and water shortages [24,41], (2) nonlinear inexact two-stage management model to optimize irrigation water allocation and (3) solution method. The details of each component are presented as follows.

### 2.1. Interval Regression Analysis Method

The benefit coefficient between unit benefit and irrigation water can be presented as a linear form whose slope is negative due to the effect of marginal utility. To better reflect actual conditions and address uncertain information, an interval regression analysis method can be introduced for obtaining inexact linear function of allocated water. A linear interval regression model can be written as) [42]:(1)$$Y\left(x\right)=A_{0}+A_{1}x_{1}+\cdots +A_{n}x_{n}=Ax$$where *x* = (1,*x*_1_,…,*x_n_*)*^t^* is a real input row vector, *t* is the symbol of transferring column vector into row vector in linear algebra, *A* = (*A*_0_,…,*A_n_*) is an interval coefficient row vector, and *Y*(*x*) is the corresponding estimated interval. In this model, interval coefficient *A_i_* is expressed as *A_i_* = (*a_i_,c_i_*) where *a_i_* means a center and *c_i_* means a radius.

In this study, interval regression analysis method based on quadratic programming is adopted because it gives more diverse spread of coefficients than a linear programming one. Meanwhile, it integrates the central tendency of least squares and possibilistic property of fuzzy regression. The goal of the interval regression based on quadratic programming is to minimize the sum of $\overset{p}{\underset{j=1}{\sum }}{\left(y_{j}-a^{T}x_{j}\right)}^{2}$ and $\overset{p}{\underset{j=1}{\sum }}c^{T}\left|x_{j}\right|{\left|x_{j}\right|}^{T}c$ according to different weights to obtain optimal interval coefficients *A_i_* = (*a_i_,c_i_*), *i* = 1,…,*n*. Therefore, the proposed model can be presented as follows:(2)$$\underset{a,c}{\min}\;J=k_{1}{\sum_{j=1}^{p}\left(y_{j}-a^{T}x_{j}\right)}^{2}+k_{2}\sum_{j=1}^{p}c^{T}\left|x_{j}\right|{\left|x_{j}\right|}^{T}c$$

The output *y_j_* should be included in the estimated output *Y*(*x_j_*), which means that the condition *y_i_* ∈ *Y*(*x_i_*), *j* = 1,2,…,*p* should be satisfied. Thus, the objective function Equation (2) should be subjected to the following constraints:(3)$$a^{T}x_{j}+c^{T}\left|x_{j}\right|\geq y_{j},j=1,\cdots ,p$$
(4)$$a^{T}x_{j}-c^{T}\left|x_{j}\right|\leq y_{j},j=1,\cdots ,p$$
(5)$$c_{i}\geq 0,i=1,\cdots ,n$$
where $\left|x_{j}\right|={\left(1,\left|x_{j1}\right|,\ldots ,\left|x_{jn}\right|\right)}^{T}$, *a* = (*a*_0_,…,*a_n_*)*^T^*, c = (c_0_,…,*c_n_*)*^T^*, *k*_1_ and *k*_2_ are weight coefficients. Generally, the larger values of *k*_1_:*k*_2_, the more central tendency would be expected. This means that the obtained interval regression results tend to be the regression results based on least squares regression method. When the values of *k*_1_ and *k*_2_ are changed, the regression results will be accordingly different. Therefore, benefit coefficients of the study can be obtained to represent inexact linear functions of the allocated water.

### 2.2. Nonlinear Inexact Two-Stage Management (NITM) Model

The NITM model can be derived from incorporating the interval parameter programming and quadratic programming methods within a two-stage stochastic programming framework (see Figure 1). Three different methods including IPP, QP and TSP have their own role and advantages in improving the agricultural water management. Specifically, because water availabilities follow a wide distribution and can’t be known ahead of time for a given time interval, pre-regulated targets must be set to guide farmers to maximize the system’s economic benefits in order to minimize the risk of system water shortages. When random uncertain parameters on the right- hand side of the model can be expressed as probability density functions and a problem is needed to analyze policy scenarios, a TSP model can be used to solve the problem [19]. However, in practical problems, the uncertainties of variables may be complex and cannot be represented as probability density functions, they may fluctuate within a certain range. To address such problems, interval parameter programming can be introduced. Moreover, uncertainty and nonlinearity may coexist in system components. This will lead to quadratic objective function and thus QP method is needed because it allows nonlinearity to exist in an objective function. In summary, the NITM model is the potential approach for dealing with above-mentioned complexities. It is written as follows:(6)$$\max\;f^{\pm }=\sum_{j=1}^{n}\left(a_{j}^{\pm }x_{j}^{\pm }+b_{j}^{\pm }\right)x_{j}^{\pm }-\sum_{j=1}^{n}\sum_{h=1}^{H}p_{h}\left(c_{j}^{\pm }y_{jh}^{\pm }+d_{j}^{\pm }\right)y_{jh}^{\pm }$$subject to:(7)$$\sum_{j=1}^{n}\left(\alpha_{j}^{\pm }x_{j}^{\pm }-y_{jh}^{\pm }\right)\leq \beta_{h}^{\pm },h=1,2,\cdots ,H$$
(8)$$x_{j}^{\pm }\geq 0,j=1,2,\cdots ,n$$
(9)$$y_{ij}^{\pm }\geq 0$$
where *f*^±^ is an objection function; $a_{j}^{\pm },b_{j}^{\pm },c_{j}^{\pm },d_{j}^{\pm }$ are the coefficients of the benefit and penalty curve (i.e., inexact linear functions), interval parameters; $x_{j}^{\pm }$ and $y_{jh}^{\pm }$ are the first-stage decision variable and second-stage decision variable in the model; $\alpha_{j}^{\pm }$ are the coefficients of left-hand side constraints; *H* is the number of the discrete value of random variables, *p_h_* is the occurrence probability of a random event, $\beta_{h}^{\pm }$ are the parameters of the right-hand side constraints. 

### 2.3. Solution Method

To solve the NITM model associated with inexact linear functions of benefits and penalties, the model should be transformed into two deterministic submodels with lower and upper bounds of the objective function values. The transformation process of the NITM model is based on interactive and derivative algorithms method [19,24,39]. Therefore, the detailed process of solution method for the model can be seen in Figure 1. Table 1 presents the related notation of the decision variables and input parameters in the developed NITM model.

## 3. Case Study 

### 3.1. Study Area

The study area is located in midstream area of the HRB in Zhangye City, Gansu Province, Northwest China (97°37′~102°06′ E, 37°44′~42°40′N and includes Ganzhou District (GZ), Linze County (LZ) and Gaotai County (GT) as shown in Figure 2. The Heihe River is divided into upstream, midstream and downstream areas by controlled hydrological stations located in Yingluoxia and Zhengyixia. The basin area is 128,000 km^2^ and it is situated in an arid region. This midstream region is composed of plains that receive an annual average rainfall level of 140 mm, and annual potential pan evaporation (E601) level reaches approximately 1400 mm. In the study area, multi-layered canals form the main channels that transfer irrigation water to the field, and as most of these canals are traditional canals with seepage losses, nearly 30% of the water is lost during water transfer [1]. Therefore, the irrigation water use efficiency coefficients of surface water and groundwater are 0.52, and 0.60, respectively [6]. There are 11 soil types in the basin, and these mainly include grey-brown desert, saline-alkali and aeolian soils [43]. More than 90% of the water consumption of the three studied administrative regions is for agricultural irrigation [26]. High water-consuming agriculture have become the dominant economic characteristic of the midstream region. The main types of crops in the study area are grain crops (denoted as GC), oil-bearing crops (OC), and economic crops (EC). 

### 3.2. Data Acquisition and Analysis

For the study area, the input information of the model basically consists of surface water and groundwater availabilities, maximum allowable water, irrigation targets and irrigation quotas, benefit and penalty coefficients of inexact linear functions. Almost all the data except for the maximum allowable water for each crop are expressed as intervals. A detailed description of these input parameters is presented in the subsections that follow.

#### 3.2.1. Irrigation Targets

Table 2 presents the irrigation targets for each crop in each subarea. Generally, different irrigation targets show the agricultural production plans determined by managers, representing the corresponding irrigation water demand as well. Irrigation water demand is the total amount of irrigation water per unit area for an entire growing period. When the water loss in a canal system and field is also summed, this value is defined as the irrigation quota (Table 2). 

Moreover, the promised irrigation targets can be obtained through multiplying the irrigated area by the irrigation quota. Irrigation quotas, which actually vary for different hydrological years, can be obtained from the Zhangye Municipal Water Conservancy Annual Report (2002–2015). An interval estimation method was used to calculate interval values with upper and lower bounds. Because the standard deviation of the collected data is unknown, a t distribution is used to estimate its range. In this study, the 95% confidence level was used to calculate the estimated interval with lower and upper bounds.

#### 3.2.2. Water Availability 

The original plan is used to decide water reallocation targets (i.e., the amount of released water to the downstream area from the Zhengyixia station) in accordance with upstream runoff inflows from the Yingluoxia station. The yearly water release to the downstream flowing though Zhengyixia station has a functional relation with runoff inflow from Yingluoxia station corresponding to different inflow levels (Figure 3). Let *p* denote the frequency for classifying each inflow level, then the classification processes are as follows: *p* ≤ 12.5% corresponds to a high inflow level, 12.5% < *p* ≤ 37.5% corresponds to a medium-high inflow level, 37.5% < *p* ≤ 62.5% corresponds to a medium inflow level, 62.5% < *p* ≤ 87.5% corresponds to a low-medium inflow level and 87.5% < *p* corresponds to a low inflow level. Then, corresponding occurrence probabilities *p_h_* of a certain inflow level can be obtained through dividing the numbers of years of each inflow level by the total numbers of years (Table 3). Moreover, the amount of released water under several typical inflow levels, namely low level, low-medium level, medium level, medium-high level and high level, are also denoted in Figure 3. Only when a gray point is greater than or equal to a black point in a straight line does it meets the original plan requirements (Figure 3). Otherwise, the requirements of the original plan are not met. It is evident that the majority of actual data cannot meet its requirements, especially under high inflow level. 

Therefore, the actual data of ecological water diversion plan was analyzed for different inflow levels, and the improved functional relationship curve is shown in Figure 3. Therefore, the best fitting relationship curve can be found and identified by least square method with minimum errors. The relationship curve can be expressed as $Q_{r}=-0.0635Q_{s}^{2}+2.8892Q_{s}-20.596$, and *R*^2^ = 0.9322 (*Q_r_*, *Q_s_* are the amount of annual inflow runoff of the Yingluoxia and Zhengyixia stations, 10^8^ m^3^). This is called the improved water diversion plan. The main difference between the original plan and improved plan lies in surface water availabilities under different inflow levels, resulting in different total water availabilities. Therefore, through the original plan and the improved plan, surface water availabilities in midstream area that presented as interval values can be quantified under different inflow levels (Table 3). Groundwater availability can be calculated based on the actual data shown in Table 3.

#### 3.2.3. Benefit and Penalty Coefficients

In this paper, the effects of marginal utility between unit benefit and the amount of irrigation water are considered. Based on related historical data of the irrigation amounts and incomes for the three crops across three subareas, it is evident that marginal utility effects are identifiable when analyzing the relationship between unit benefit and the amount of irrigation water. Marginal utility refers to a reduction in unit benefit of each unit of irrigation water as the slope of the benefit curve for allocated water is negative. Benefits and penalties are expressed as inexact linear functions of the allocated water and shortage, respectively. According to Tanaka and Lee [42], in the case of *k*_1_ >> *k*_2_, the obtained center regression line tends to be the regression line obtained by traditional method like least squares method. Therefore, in this study, the results of *k_1_* = 1, *k*_2_ = 0.0001 are considered as the benefit coefficients of the three crops because they have more central tendency. It is difficult to construct an accurate penalty coefficients, i.e., loss function [44], the conceptual nature of the penalty functions (e.g., two-sided or one-sided, slope, nonlinearity) may vary in different applications. Penalty coefficients are obtained based on the reference from Huang et al. [24] and thus the inexact linear functions of water shortages can be determined (Table 4).

### 3.3. Nonlinear Inexact Two-Stage Management Model for Optimal Agricultural Water Allocation

In this study, managers are responsible for allocating water to three crops in three irrigation subareas. They need to make a decision regarding the irrigation target (first-stage) for each subarea and each crop when the upcoming inflow is unknown, so farmers can arrange their crop production activities accordingly. If the promised amount is delivered, it will bring benefits to the farmers; however, if the promised water is not delivered, it may result in penalties on the local economy. Subsequently, when the inflow is known, a second-stage decision has to be made to adjust the predefined decision and minimize the penalties due to any infeasibility. Moreover, the impacts of marginal utility are desired to reflect because the benefit and penalty are expressed as inexact linear functions of allocated water and shortage. When the objective function denotes a maximum economic benefit, it is a nonlinear problem [38]. Therefore, the problem can be formulated as the NITM model.

Objective function:(10)$$\max\;f^{\pm }=\sum_{i=1}^{I}\sum_{j=1}^{J}\left(a_{ij}^{\pm }W_{ij}^{\pm }+b_{ij}^{\pm }\right)W_{ij}^{\pm }-\sum_{i=1}^{I}\sum_{j=1}^{J}\sum_{h=1}^{H}p_{h}\left(c_{ij}^{\pm }S_{ijh}^{\pm }+d_{ij}^{\pm }\right)S_{ijh}^{\pm }$$where $\sum_{i=1}^{I}\sum_{j=1}^{J}\left(a_{ij}^{\pm }W_{ij}^{\pm }+b_{ij}^{\pm }\right)W_{ij}^{\pm }$ are the economic benefits when the promised water for irrigation target is delivered and $\sum_{i=1}^{I}\sum_{j=1}^{J}\sum_{h=1}^{H}p_{h}\left(c_{ij}^{\pm }S_{ijh}^{\pm }+d_{ij}^{\pm }\right)S_{ijh}^{\pm }$ are the economic penalties when the promised water is not delivered and there are water shortages.

Constraints:(11)$$\sum_{i=1}^{I}\sum_{j=1}^{I}\left(W_{ij}^{\pm }-S_{ijh}^{\pm }\right)\leq Q_{h}^{\pm }{}_{sw}\eta_{sw}\eta_{ag}+Q_{h}^{\pm }{}_{gw}\eta_{gw},\forall h$$

Equation (11) is the water availability constraint. The total water demand is less than total water availability under different levels *h*, including surface water $Q_{h\;sw}^{\pm }$ and groundwater $Q_{h\;gw}^{\pm }$ levels.

(12)$$W_{ij\max}\geq W_{ij}^{\pm }\geq S_{ijh}^{\pm },\forall i,j,h$$

Equation (12) is the maximum allowable water constraint.

(13)$$S_{ijh}^{\pm }\geq 0,\;\forall i,j,h$$

Equation (13) is a non-negative water irrigation target deficit constraint.

In this study, penalty coefficients of $c_{ij}^{+}$ and $d_{ij}^{+}$, which are related to the second stage decision variable $S_{ijh}^{\pm }$, have the same sign (i.e., $c_{ij}^{+}$ > 0 and $d_{ij}^{+}$ > 0). According to solution method listed above (see Figure 1), all $S_{ijh}^{-}$ values corresponding to f^+^, and all $S_{ijh}^{+}$ values corresponding to *f*^--^. As the objective function maximizes overall economic benefits, thus submodel corresponding to the upper bound is obtained first. Let $W_{ij}^{\pm }=W_{ij}^{-}+\Delta W_{ij}z_{ij}$ by introducing decision variables *z_ij_*, where $\Delta W_{ij}=W_{ij}^{+}-W_{ij}^{-}$ and *z_ij_* ∈ [0,1]. Therefore, optimal water allocation solutions are as follows:(14)$$W_{ij,opt}^{\pm }=W_{ij}^{-}+\Delta W_{ij}z_{ij}{}_{,opt}$$

(15)$$S_{ijh,}^{\pm }{}_{opt}=\left[S_{ijh}^{-}{}_{,opt},\;S_{ijh,}^{+}{}_{opt}\right]$$

(16)$$f_{opt}^{\pm }=\left[f_{opt}^{-},\;f_{opt}^{+}\right]$$

Actual irrigation targets:(17)$$AW_{ijh}^{\pm }{}_{,opt}=W_{ij,opt}^{\pm }-S_{ijh,}^{\pm }{}_{opt}$$

According to regional agricultural development requirements, the total agricultural irrigated area must remain unchanged, and water consumption must be reduced in the midstream region to allow more water to be released to the downstream area to support the ecological safety. Therefore, the promised total water consumption can be regarded as a major factor in policy scenario analysis. Four scenarios are examined as follows:Scenario 1:irrigation targets maintain existing conditions (i.e., $W_{ij}^{\pm }$), and the sum of all irrigation targets is the total water consumption in midstream area. Scenario 2:irrigation targets are reduced by 10%, representing 90% of the current level (i.e., $0.9W_{ij}^{\pm }$). Scenario 3:irrigation targets are reduced by 20%, representing 80% of the current level (i.e., $0.8W_{ij}^{\pm }$). This scenario denotes that the total water consumption is reduced by 20%.Scenario 4:irrigation targets are reduced by 30%, representing 70% of the current level (i.e., $0.7W_{ij}^{\pm }$). This scenario represents the case that the total water consumption is reduced by 30%.

## 4. Results Analysis and Discussion

### 4.1. Optimal Irrigation Water Allocation

The NITM model can effectively tackle uncertainties expressed as discrete intervals and probability distributions. Optimal solutions can reflect a tradeoff between economic benefits and related pre-regulated policy targets, and reflect the effects of marginal utility between unit benefit and irrigation water amount. When water availabilities do not satisfy irrigation targets, it will lead to water shortages and reduction of economic benefits (i.e., penalties due to infeasibilities) and further system-failure risks. The obtained solutions are expressed as deterministic or interval values. Generally, solutions expressed as intervals show that the decision variables are sensitive to uncertainties in the model [20]. The interval solutions can provide many decision solutions for a fluctuating range and further investigate the relationship between economic benefits and violations of irrigation target risks. More practically, the greater the width of upper bound and lower bound, the greater degree of uncertainty. The smaller the width of upper bound and lower bound, the more reliable and accurate decision-making becomes. 

Figure 4a–d show optimal solutions for water allocation targets of the NITM model based on the improved plan (similar results can be obtained from the original plan). This result shows the improved water allocation to avoid unreasonable solutions from the original plan. The NITM model is capable of analyzing various policy scenarios for different irrigation targets. Water shortages occur when water availabilities do not satisfy irrigation targets. From scenarios 1 to 4, due to reductions of total water consumption, promised irrigation targets and optimal water allocation targets are diminishing accordingly, causing system losses and system-failure risk of water shortages to decline as well. This result also implies that different irrigation targets corresponds to different water shortages or surpluses, thereby affecting economic benefits and system-failure risks. Moreover, the optimal solutions for OC (i.e., oil-bearing crops) in the GZ would be [350.4, 380.5] × 10^4^ m^3^, [357.8, 376.1] × 10^4^ m^3^, [359.2, 381.5] × 10^4^ m^3^, [376.4, 391.9] × 10^4^ m^3^, and [402.4, 420.6] × 10^4^ m^3^ under inflow levels from *h* = 1 to *h* = 5, respectively. The optimized water allocation for EC (i.e., economic crops) in the GZ would be [54938.0, 55983.9] × 10^4^ m^3^, [54791.4, 56179.7] × 10^4^ m^3^, [54971.8, 56216.4] × 10^4^ m^3^, [55321.4, 56669.2] × 10^4^ m^3^, and [55671.0, 57835.7] × 10^4^ m^3^ under inflow levels from *h* = 1 to *h* = 5, respectively. Their absolute values of difference between upper bound and lower bound are 1045.9, 1388.3, 1244.6, 1347.8, 2164.7 × 10^4^ m^3^, respectively. More importantly, the amount of these irrigation waters is also important for agricultural development and decision making in arid/semiarid regions. Because the results of water allocation are presented as gross irrigation water, thus, according to the model solution in the GZ, i.e., the amount of irrigation water loss depending on the water conveyance efficiency is [18391.4, 19852.4] × 10^4^ m^3^, the total amount of water loss is [25747.9, 27793.3] × 10^4^ m^3^, and the amount of water loss depending on the irrigation efficiency is [6356.5, 7940.9] × 10^4^ m^3^. Similar results can be obtained in other subareas, according to the water conveyance efficiency and irrigation efficiency. 

Similarly, optimal solutions for EC in the GT would be [9783.2, 15922.7] × 10^4^ m^3^, [9578.7, 16070.5] × 10^4^ m^3^, [9830.4, 16098.2] × 10^4^ m^3^, [10318.0, 16439.8] × 10^4^ m^3^, and [10805.6, 17319.9] × 10^4^ m^3^ under the inflow levels from *h* = 1 to *h* = 5, respectively. According to $AW_{ij,opt}^{\pm }=W_{ij}^{-}+\Delta W_{ij}z_{ij,opt}-S_{ijh,opt}^{\pm }$, optimal targets would be identified by decision variables $z_{ij,opt}$. For example, in the initial scenario (S1), $z_{13,opt}=z_{23,opt}=1$, indicating that the optimized irrigation targets reach their upper bounds and no water shortage should occur. This result also shows that planners hold positive attitudes to water availabilities for the EC in the GZ and LZ. Other decision variables ($z_{ij,opt}=0$) approach their lower bounds and express conservative attitudes. Under such circumstances, water shortages $S_{ijh,opt}^{\pm }$ are expressed as $\Delta W_{ij}\left(1-z_{ij,opt}\right)$. As the decision variables $z_{ij,opt}$ are in the range 0–1, thus the satisfaction level of irrigation water requirements for different crops can be measured from $z_{ij,opt}$. Therefore, resourse decisions can obtain improved overall solutions by realigning the first-stage decision with possible realizations of random uncertainty. On the one hand, if the promised water is not delivered, a second-stage decision can be made to adapt to irrigation target and thus minimize the system penalties because of infeasibility. On the other hand, if the runoff inflow is at high level, then, it may cause more waste of irrigation water. This is because a conservative irrigation target lead to irrigation water cannot be fully utilized. Therefore, the results reflect tradeoffs between irrigation targets and random runoff inflow.

The results show that scenarios 2, 3, and 4 present the same change trend as that of scenario 1 under different inflow levels associated with the total water availabilities. However, among the four scenarios, solutions present different patterns of optimal water allocation for subarea *i* to crop *j* under a certain level. For example, under *h* = 1 level, optimized water allocation for EC in the GZ would be [54938.0, 55983.9] × 10^4^ m^3^, [49189.9, 50432.6] × 10^4^ m^3^, [44790.0, 45783.7] × 10^4^ m^3^, and [41246.3, 41924.1] × 10^4^ m^3^ in scenarios 1–4, respectively. Therefore, among four scenarios, variation trends for both lower and upper bounds under a certain inflow level are the same (scenario 1 > scenario 2 > scenario 3 > scenario 4). Rather, both upper and lower bounds of optimized water allocation present significantly decreasing trends. However, there are also increasing in the lower bound values while decreasing in the upper bound values in the four scenarios (i.e., GT-GC, LZ-EC). This result indicates that interval variations and mid-values of scenario 1 are larger than others, verifying the effects of marginal utility of the objective function. Moreover, there are also increasing lower and upper bound interval values in four scenarios (i.e., LZ-GC, LZ-OC, and GT-EC). When lower bound values of optimal irrigation water allocations are equal to upper bound values, the system arrives at a minimum marginal utility and approaches maximum system benefits.

### 4.2. Economic Benefits Analysis

Figure 5 presents a system economic benefits comparison based on the original plan and improved plan for different scenarios. Different agricultural irrigation management policies associated with various irrigation water targets and water availabilities bring about different system economic benefits. In terms of the improved plan and original plan, system benefits of the improved plan are greater than those of the original plan for different scenarios. The higher level of water availability permitted for right-hand side constraints generates a loose constraint and thus expands the decision space. The expected system economic benefits are denoted as interval values $f_{opt}^{-}$ and $f_{opt}^{+}$, indicating that benefits can fluctuate over a certain range, and thus decision alternatives can be generated. When objection function values approach their lower bounds, fewer system benefits can result while water shortages are lower corresponding to a lower risk of promised target violations. Otherwise, a higher system benefit can be obtained when water demands are satisfied, but water shortage levels may increase, and thus greater system failure risks can result. Therefore, the results can reflect interrelationships among promised irrigation targets, system economic benefits and risk levels.

Take the improved plan as an example to describe system economic benefits for four scenarios. The gross system benefits from irrigation water would be [2.36, 4.75] × 10^9^ Yuan, [2.25, 4.21] × 10^9^ Yuan, [2.14, 3.77] × 10^9^ Yuan, and [1.99, 3.39] × 10^9^ Yuan, respectively. Because the promised irrigation targets are reducing from scenarios 1 to 4, fewer system benefits would be achieved associated with a lower risk of violating water allocation constraints. The above analyses show that different policies on irrigation targets will lead to different system economic benefits and risk levels.

### 4.3. Discussion

The main difference between the original plan and improved plan pertains to surface water availability under different inflow levels, which gives rise to different total water availabilities. Therefore, irrigation targets are progressively lower from scenarios 1 to 4. Only the lower bound of the total available water volume under *h* = 2 based on the IP is lower than that of the original plan, but in other cases, both lower and upper bounds of the total available water based on the improved plan are larger than those of the original plan. Figure 6a–c present the results for optimal water allocation under different scenarios based on the improved plan and original plan. 

The results obtained from the improved plan are greater than or equal to those of the original plan. For example, for *h* = 1, the total optimized water allocation is [79918.4, 100681.6] × 10^4^ m^3^ and [78069.7, 98730.3] × 10^4^ m^3^ in scenario 1, [80468.9, 100131.1] × 10^4^ m^3^ and [78388.4, 98411.6] × 10^4^ m^3^ in scenario 2, [82851.7, 97748.3] × 10^4^ m^3^ and [80742.0, 96058.0] × 10^4^ m^3^ in scenario 3, [84000.0, 96600.0] × 10^4^ m^3^ and [82050.3, 94749.7] × 10^4^ m^3^ in scenario 4. With the total water consumption reducing, the lower bound of total optimal water allocation increases, while the upper bound values decrease. This result indicates that the interval range becomes much narrower, demonstrating the existence of marginal utility in the objective function. This result can also provide many decision alternatives for further analysis on the relationship between economic benefits and violations of irrigation target risks. Therefore, the improved plan corresponds to a lower water shortage level than the original plan under the same inflow level, and thus leads to less system-failure risk.

For a certain crop (taking EC in GZ as an example), optimal water allocation would be [54938.0, 55983.9] × 10^4^ m^3^ and [54723.7, 55751.4] × 10^4^ m^3^ in scenario 1; [49189.9, 50432.6] × 10^4^ m^3^ and [48975.6, 50203.3] × 10^4^ m^3^ in scenario 2; [44790.0, 45783.7] × 10^4^ m^3^ and [44536.9, 45593.3] × 10^4^ m^3^ in scenario 3; [41246.3, 41924.1] × 10^4^ m^3^ and [41188.7, 41880.5] × 10^4^ m^3^ in scenario 4. This result indicates that the former has more total available water, thus expanding decision-making space on the right-hand side constraints. For scenarios 1 to 4, optimal solutions decrease both lower and upper bounds values in the GZ-EC. However, in the LZ-GC, GT-GC and GT-EC, optimized solutions gradually increase both lower and upper bound values. 

Generally, if more water is provided for agricultural irrigation, it will improve the crop yield and thus increase system benefits. More importantly, for different scenarios, the rate of economic growth based on the improved plan and original plan are not the same. For example, in terms of upper bound of economic benefits, the rate of growth based on the improved plan between scenario 1 and scenario 2, scenario 2 and scenario 3, scenario 3and scenario 4 are 12.7%, 11.9% and 11.1%, respectively, which is greater than 11.8%, 11.2% and 10.4% based on the original plan. Therefore, the NITM model can reflect how to achieve greater economic benefits in the case of considering the increase of water availabilities.

## 5. Conclusions

In this study, a nonlinear inexact two-stage management (NITM) model is formulated, which incorporates methods of interval parameter programming (IPP) and quadratic programming (QP) into the two-stage stochastic programming (TSP) optimization framework to facilitate optimal agricultural irrigation water allocation. The model can deal with uncertainties not only discrete intervals, but also in probability distributions, and nonlinear problems in objective functions simultaneously. The effects of marginal utility are considered when limited water resources are allocated to multiple water users. Hence, relationships between benefits and allocated water, and between penalties and water shortages are expressed as inexact linear functions form in the model, which is an improvement for general TSP framework where its benefit coefficients are presented as deterministic values. The results from four scenarios with different irrigation targets can reflect interrelationships among promised irrigation targets, system economic benefits and risks. Meanwhile, an improved water diversion plan is obtained, which can overcome difficulties associated with fulfilling released water requirements under higher inflow levels. Therefore, optimal water allocation solutions can be obtained based on the original plan and improved plan, and careful comparisons indicate that the improved plan is superior to the original plan in terms of practical operations and economic benefits.

Furthermore, the model has demonstrated a strong practical applicability in the agricultural water management of midstream area of the HRB. The model can deal with the impacts of marginal utility and uncertainties that commonly exist in the agricultural water management problems with limited irrigation water. Although marginal values of water was calculated based on such a site-specific data, the results suggest that its calculation method and concept are applicable to different practical problems. Although optimal solutions of sound management policies have been obtained using the NITM model, still some further research works should be conducted. More system conditions should be taken into consideration (e.g., soil moisture, soil texture, crop yields, irrigation water resource allocation in canal systems and deficit irrigation levels). Therefore, how to address these complexities including more natural condition factors strategically and the subjective nature of computing the marginal utilities will be an interesting topic in the future studies.

## Figures and Tables

**Figure 1 ijerph-16-01884-f001:**
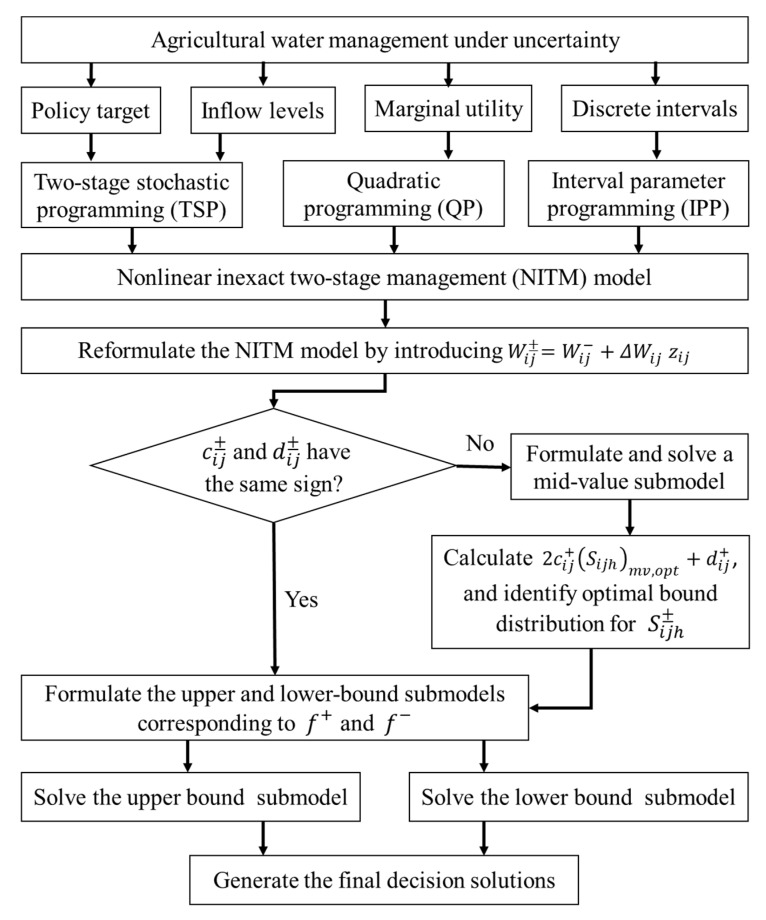
The framework of the NITM model.

**Figure 2 ijerph-16-01884-f002:**
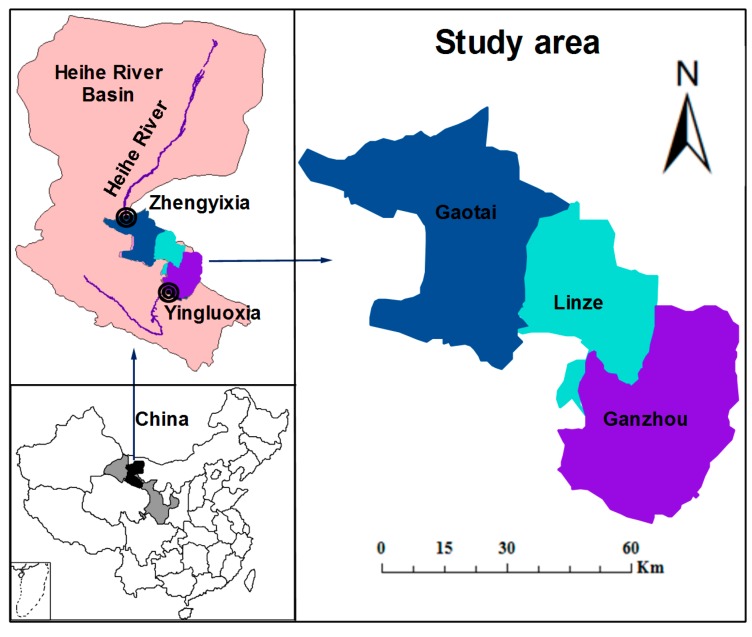
The study area.

**Figure 3 ijerph-16-01884-f003:**
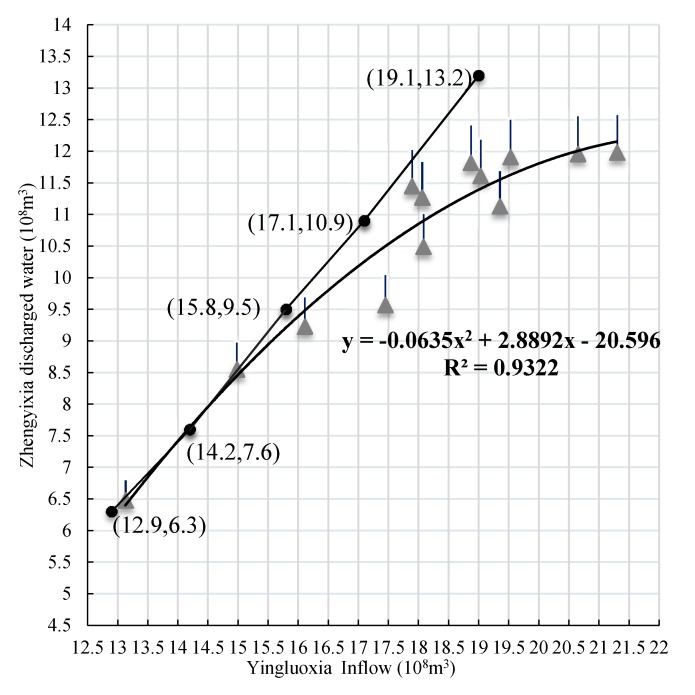
The relationship curve for water reallocation targets of the ‘97’ water diversion plan and the improved plan. Note: The straight line is the runoff relationship of the original plan. The black solid points are the reallocation targets of the original plan corresponding to different Yingluoxia inflow levels and the gray solid points are the actual data allowing for the presence of a 5% relative error.

**Figure 4 ijerph-16-01884-f004:**
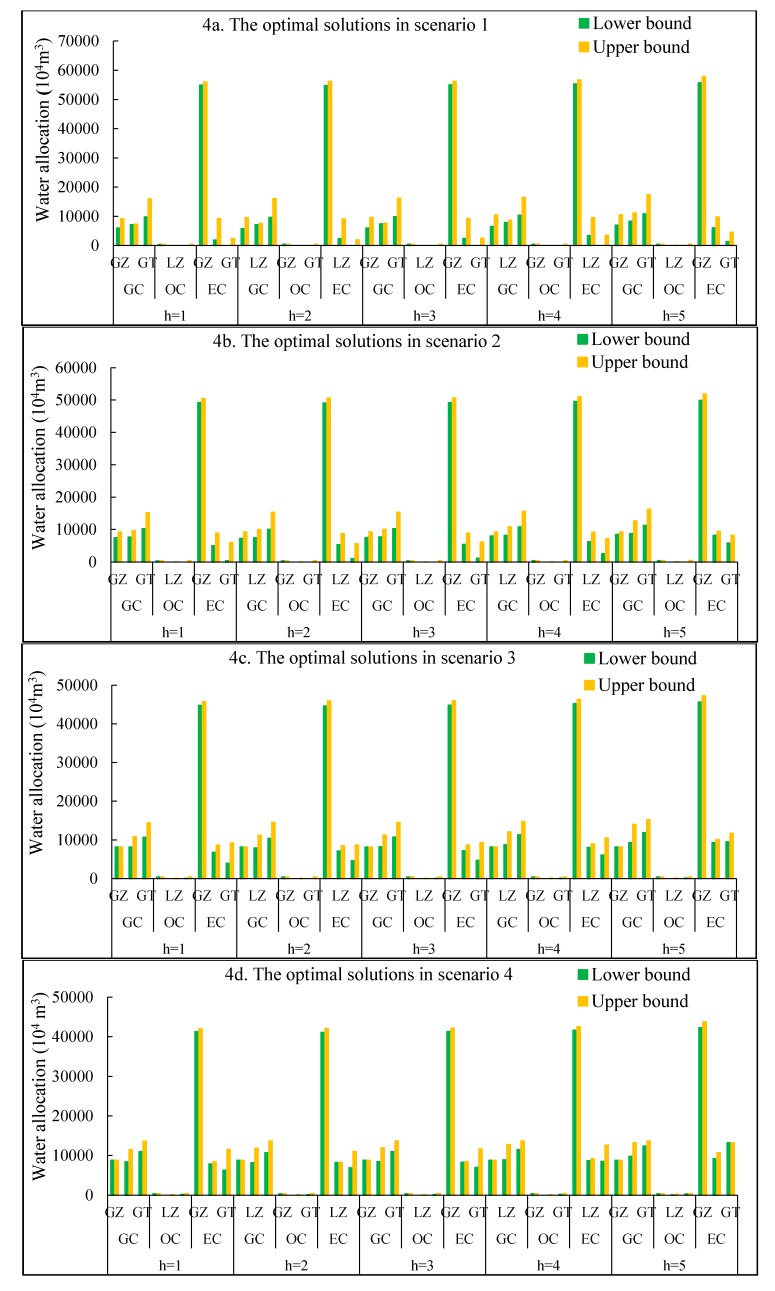
Comparison of results in different scenarios and different inflow levels based on the improved plan. (**a**) The optimal solutions in scenario 1; (**b**) The optimal solutions in scenario 2; (**c**) The optimal solutions in scenario 3; (**d**) The optimal solutions in scenario 4.

**Figure 5 ijerph-16-01884-f005:**
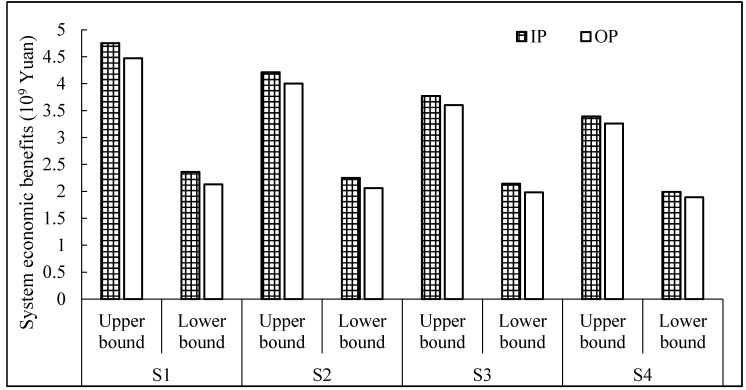
Economic benefits comparison based on the original plan (OP) and improved plan (IP) under different scenarios.

**Figure 6 ijerph-16-01884-f006:**
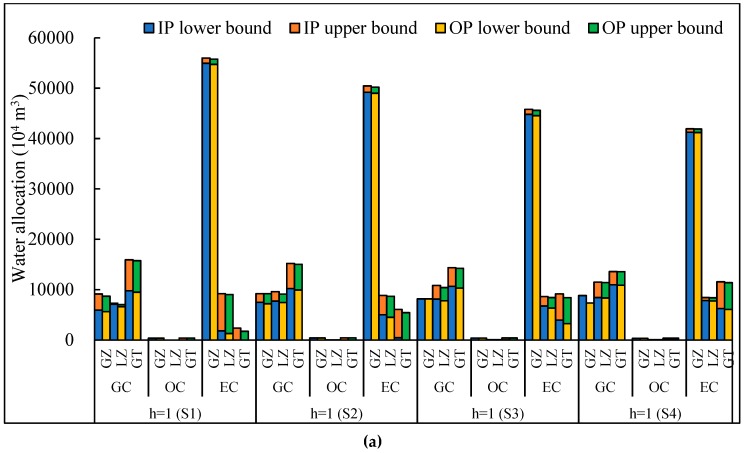
Comparison under different scenarios based on the original plan (OP) and improved plan (IP) (**a–c**).

**Table 1 ijerph-16-01884-t001:** Notation of the decision variables and input parameters.

Symbol	Notation
*f^±^*	System economic benefits (10^8^ Yuan)
*i*	Subarea, *i =* 1,2,3. *I =* 3 (*i =* 1 denotes GZ; *i =* 2 denotes LZ and *i =* 3 denotes GT)
*j*	Crop, *j =* 1,2,3. *J* = 3 (*j =* 1 denotes GC; *j =* 2 denotes OC and *j =* 3 denotes EC)
*h*	Inflow level, *h =* 1,2,3,4,5. *H* = 5 (*h =* 1, low level; *h =* 2, low-medium level; *h =* 3, medium level; *h =* 4, medium-high level and *h =* 5, high level)
*p_h_*	Probability of inflow level *h*
$B_{ij}^{\pm }=a_{ij}^{\pm }W_{ij}^{\pm }+b_{ij}^{\pm }$	The benefit coefficient, benefit for subarea *i* to *j* crop of promised allocated irrigation water (10^4^ Yuan/10^4^ m^3^)
$a_{ij}^{\pm }$	The slope of the relationship curve between unit benefit and irrigation water amount for subarea *i* to *j* crop ($a_{ij}^{\pm }$ < 0 denotes that unit benefit decrease with irrigation water increase)
$b_{ij}^{\pm }$	The intercept of the relationship curve between unit benefit and the amount of irrigation water for subarea *i* to *j* crop ($b_{ij}^{\pm }\;$> 0)
$C_{ij}^{\pm }=c_{ij}^{\pm }S_{ijh}^{\pm }+d_{ij}^{\pm }$	The penalty coefficient, reductions/penalties for water users to subarea *i* for crop *j* of water not irrigated (10^4^ Yuan/10^4^ m^3^), $C_{ij}^{\pm }>\;B_{ij}^{\pm }$
$c_{ij}^{\pm }$	The slope of the relationship curve between unit penalty and the amount of irrigation water shortages for subarea *i* to *j* crop ($c_{ij}^{\pm }>0$ denotes that as water shortages increase, penalties also increase)
$d_{ij}^{\pm }$	The intercept of the relationship curve between unit penalty and the amount of irrigation water shortages for subarea *i* to *j* crop ($d_{ij}^{\pm }>0$)
$W_{ij}^{\pm }$	Pre-regulated irrigation targets promised to water users for subarea *i* to *j* crop. This is the first stage decision variable (10^4^ m^3^)
$Q_{h\;sw}^{\pm }$	Random variable of surface water availability for irrigation under a certain inflow level *h* (10^4^ m^3^)
$Q_{h\;gw}^{\pm }$	Random variable of groundwater availability for irrigation under a certain inflow level *h* (10^4^ m^3^)
*η_sw_*	Irrigation water use efficiency coefficient of surface water, 0.52
*η_gw_*	Irrigation water use efficiency coefficient of groundwater, 0.6
*η_ag_*	The proportion of agricultural irrigation in three studied administrative regions to water consumption, 0.9
$S_{ijh}^{\pm }$	Water shortages for subarea *i* to *j* crop under inflow level *h*, which is the amount of water that does not satisfy the promised target under the level of total water availabilities (10^4^ m^3^)
*W_ij_* _max_	The maximum allowable water for subarea *i* to *j* crop (10^4^ m^3^)
*z_ij_*	Decision variable to solve the NITM model, $W_{ij}^{\pm }=W_{ij}^{-}+\Delta W_{ij}z_{ij}$, $\Delta W_{ij}=W_{ij}^{+}-W_{ij}^{-}$ and $z_{ij}\in \left[0,\;1\right]$

**Table 2 ijerph-16-01884-t002:** Irrigation targets, irrigation quota and the maximum allowable water.

Subarea	Irrigation Targets (10^8^ m^3^)		
	GC	OC	EC
GZ	[1.05059, 1.2598]	[0.04794, 0.07347]	[5.45679, 6.24537]
LZ	[1.88783, 2.14500]	[0.03234, 0.05051]	[1.35234, 1.52921]
GT	[1.95183, 2.14411]	[0.05304, 0.06544]	[1.89169, 2.15210]
Irrigation quota for each crop (m^3^/ha)			
GZ	[5595, 7020]	[4320, 5625]	[11232, 12355]
LZ	[7500, 9720]	[5175, 6750]	[11659, 12825]
GT	[6975, 9840]	[4725, 6150]	[12105, 13316]
Maximum allowable water (10^8^ m^3^)			
GZ	1.3385	0.0767	6.3192
LZ	2.1155	0.0527	1.6083
GT	2.3662	0.0670	2.2892

**Table 3 ijerph-16-01884-t003:** Water availabilities for irrigation under different inflow levels (10^8^ m^3^).

Inflow Level	Probability	Surface Available Water from the IP	Surface Available Water from the OP	Allowable Groundwater	Total Available Water from the IP	Total Available Water from the OP
Low	0.12	6.79	6.60	[2.20, 2.28]	[8.99, 9.07]	[8.80, 8.88]
Low-medium	0.25	[6.57, 6.79]	6.60	[2.29, 2.44]	[8.86, 9.23]	[8.89, 9.04]
Medium	0.32	[6.57, 6.60]	6.30	[2.45, 2.66]	[9.02, 9.26]	[8.75, 8.96]
Medium-high	0.17	[6.60, 6.86]	6.20	[2.67, 2.77]	[9.27, 9.63]	[8.87, 8.97]
High	0.14	[6.86, 7.62]	5.80	[2.78, 2.88]	[9.64, 10.5]	[8.58, 8.68]

Note: IP denotes the improved plan while OP denotes the original water diversion plan.

**Table 4 ijerph-16-01884-t004:** The coefficients of benefit and penalty curve based on interval regression analysis (Yuan/10^4^ m^3^).

Subarea	Upper Bound			Lower Bound		
	GC	OC	EC	GC	OC	EC
Benefit coefficients when water demand is satisfied ($B_{ij}^{\pm }=a_{ij}^{\pm }W_{ij}^{\pm }+b_{j}^{\pm }$)						
GZ	−1.0264*x* + 55955	−36.1470*x* + 54614	−0.2572*x* + 56610	−1.0264*x* + 44890	−36.1470*x* + 26294	−0.2572*x* + 25803
LZ	−0.8515*x* + 41516	−13.7720*x* + 22893	−1.1245*x* + 40698	−0.8515*x* + 29228	−13.7720*x* + 15844	−1.3427*x* + 27671
GT	−0.4149*x* + 32466	−26.0970*x* + 42300	−0.7311*x* + 49580	−0.4149*x* + 27408	−27.8260*x* + 24563	−0.7311*x* + 26426
Penalty coefficients when water demand is not satisfied ($C_{ij}^{\pm }=c_{ij}^{\pm }S_{ijh}^{\pm }+d_{ij}^{\pm }$)						
GZ	2.0528*x* + 95124	72.2940*x* + 92844	0.5144*x* + 96237	1.8475*x* + 68682	65.0646*x* + 40230	0.4630*x* + 39479
LZ	1.7030*x* + 70577	27.5440*x* + 38918	2.2490*x* + 69187	1.5327*x* + 44719	24.7896*x* + 24241	2.4169*x* + 42337
GT	0.8298*x* + 55192	52.1940*x* + 71910	1.4622*x* + 84286	0.7468*x* + 41934	50.0868*x* + 37581	1.3160*x* + 40432

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
