# Peer review of "A Nonlinear Inexact Two-Stage Management Model for Agricultural Water Allocation under Uncertainty Based on the Heihe River Water Diversion Plan"

_ijerph, 2019, doi:10.3390/ijerph16111884_

Round 1

Reviewer 1 Report

This paper demonstrates an agricultural water optimization model that introduces stochastic elements and economic precepts to compensate for uncertainty and to maximize utility under changing conditions. The authors have done a great job contextualizing this study with research from optimization modeling. However, there is very little background regarding the economics side, which is a large part of the novelty of this paper. Hydroeconomics modeling has a rich literature, and it should be considered in the introduction. The methods section is also pretty thin regarding data collection and estimation of marginal utility. It is written as a case study, but could be shifted slightly to be more generalized, which I believe would be an improvement. The math in particular is presented very completely, but the omission of some details, such as data sources, would nevertheless make it difficult to replicate in another area. It would also be of greater interest to people outside the field of optimization if all of the formulas were very explicit in terms of their conceptual meaning. Overall, it is an interesting and well-written paper, and is publishable with a few changes. 

Please find the attached .pdf with a few minor comments, but primarily some language editing. 

Recommendations

Change the title wording and punctuation to the following: A nonlinear inexact two-stage management model for agricultural water allocation under uncertainty based on the Heihe River water diversion plan

Consider getting rid of the acronyms and just writing out the words instead. There is no compelling reason to write OP instead of original plan, the acronym IMP is introduced and never used, and a phrase like “This model incorporates IPP and QP into TSP” is very difficult to parse.

This paper does a great job of placing the current study within the context of previous optimization models, but by concentrating on the marginal utility of water, it also becomes a hydroeconomic model. The hydroeconomic literature is very rich. Please add a paragraph to the introduction to contextualize this paper with respect to previous hydroeconomics models. I recommend the following review paper as a starting point:

Harou, J.J., Pulido-Velazquez, M., Rosenberg, D.E., Medellín-Azuara, J., Lund, J.R. and Howitt, R.E., 2009. Hydro-economic models: Concepts, design, applications, and future prospects. Journal of Hydrology, 375(3-4), pp.627-643.

This would strengthen your paper by, for example, allowing you to address elasticity, and how it relates to your methods of calculating utility. For an ecological system, there may be a threshold of flow needed to support a given species. In that case, the ecological need for water is highly inelastic. The elasticity of agricultural water needs can change within a season, depending on the plant phenological stage.

Section 2.1: Great job talking the reader through these equations, I like how this is formatted. There is one component of the equations that is not addressed: the indexing. It is not obvious whether t or T stands for time, and whether instances from j=1 to p are representative of time steps or different components of the system such as agricultural fields, or agriculture versus river ecology etc. It may be possible to figure out based on the context, but it would be preferable to clarify this for the reader.

Line 172: “water availabilities are random” It would be more accurate to say that water availabilities follow a wide distribution and can’t be known ahead of time for a given time interval.

Line 194: I recommend moving the explanation of the + and – notation closer to the beginning of the paper, after line 104, where this notation is first used.

Figure 1 is not sharp, it will need a higher quality file for the final publication. More importantly, it uses notation that is not introduced in the text. What are W, S, and z? Both the steps of the analysis and the notation need to be consistent with the main text. *Note: These are defined after line 317. See comment below regarding a table of parameters to aid the reader.

Figure 2 text is too small to read comfortably.

Line 240 “can be obtained from statistical data” Please add details regarding the exact source of the underlying data: was it from the government, or from producers, or …? What statistical methods were used to prepare it for the main analysis? Please add enough information to make the study plausibly reproducible.

Please move Figure 3 closer to lines 255-256, this is very interesting but only makes sense in the context of the figure.

If there is any possible way to put the numbers in Tables 1 and 2 into the same units, it would be preferable – it would make it much easier to compare the values. Either 10^4 or 10^8 would be fine.

Table 3: it is not clear what the relationship is between the formulas introduced in this table and the rest of the formulas in the methods, this should be addressed along with the comments on Figure 1.

I recommend creating a table of parameters and their meanings (and preferably their units), to accompany Figure 1. This will make the equations much more legible for the reader.

Figure 4 is a bit hard to read – would it be possible to print the graphs sideways on the page?

Line 464: Are the system benefits equivalent to gross profits from irrigation water?

Figure 6: same note as figure 4. Also, the lines and dots are a bit hard to interpret, and the lines don’t make theoretical sense, because there’s no actual continuity of the underlying data. Please use whiskers to indicate upper and lower bound intervals, rather than having the lines and dots.

Author Response

We are grateful to Reviewer’s insightful review. The provided comments have contributed substantially to improving the paper. Based on them, we have made significant efforts to revise the paper, the following details are presented.

Reviewer 2 Report

This manuscript by Zhang et al proposed a model for optimal agricultural irrigation water management problems under uncertainty. This work would of interest for local goverments and researchers. Work is well written, but did not escape a few mistakes and oversights.

The proposed model allows error propagation or prediction accuracy assessment. Whether the results were compared with the results of the models used so far.

How the irrigation target was determined on the basis of historical data or predictions - this requires clarification.

Tables presenting the results are very unreadable. Full redundant abbreviations and values in brackets that are not sufficiently described

Author Response

We are grateful to Reviewer’s insightful review. The provided comments have contributed to improving the paper. Based on them, we have made efforts to revise the paper, the following details are presented.

Reviewer 3 Report

I think the premise of the paper is interesting, and the methods are novel. However, I found it confusing to weave all the equations throughout the text. I thought that made it very difficult to understand conceptually what the authors were doing and locate high-level information. 

Some very important information was mentioned only tangentially or not at all. For example, I couldn't figure out how the authors actually derived site-specific relative weights for various uses. I think they referred to another document, but I'm not sure.

Overall, it would be much easier to understand if the authors used text space for narrative description in language that is as ACCESSIBLE as possible to the broadest audience and then put equations (because there are many of them) in tables with descriptions, in the main text if need be or in the SI. Currently, it is not easy to understand the high-level information in the paper, especially for people who are not experts in the methods used by the authors. I believe this point is extremely important because the authors present policy-relevant analysis that, ideally, will be read by people who may have extremely limited background in optimization, water quality modeling, etc.

There were various English issues that should be addressed, for example (not exhaustive):

o  “in arid region of northwest China” (in THE arid region? In AN arid region?)

o   “Especially, midstream region” (THE midstream region)

o  “full of competitions” – I assume you mean that there are lots of competing demands?

o  “dramatically conflicts” – do you mean dramatic conflicts?

Author Response

We are grateful to Reviewer’s insightful review. The provided comments have contributed substantially to improving the paper. Based on them, we have made efforts to revise the paper, the following details are presented.

Round 2

Reviewer 2 Report

Thank you very much to the author for taking into account my comments.

Author Response

We are grateful to Reviewer’s insightful review.

Reviewer 3 Report

The authors have done a good job summarizing the methods in a table. I am still wondering how site-specific data was used to calculate marginal values of water in different applications. The authors amply describe interval regression analysis in their response and in the text. But what data are these being regressed against? It would seem like determining the relative value of upstream vs. downstream uses of water would embed many subjective factors. What is the basis for this? Wouldn't it depend on alternatives? For example, if farmers could extract less river water without compromising crop yields, the marginal value would be zero, right? This would seem to depend on crop yield models, and then the importance of yields would in turn depend on how important local farms are to the overall regional food system. None of this is really explained, and that seems to be a big gap. 

Author Response

We are grateful to Reviewer’s insightful review. The provided comments have contributed to improving the paper.
